# The adverse effects of bisphosphonates in breast cancer: A systematic review and network meta-analysis

**Christopher Jackson**[1‡]**, Alexandra L. J. Freeman** [2‡*]**, Zśofia Szlamka**[2]**, David J. Spiegelhalter**[2]

**1** MRC Biostatistics Unit, University of Cambridge, Cambridge, United Kingdom, **2** Department of Pure Mathematics & Mathematical Statistics, Winton Centre for Risk & Evidence Communication, University of Cambridge, Cambridge, United Kingdom

‡ These authors are joint senior authors on this work
* Alex.freeman@maths.cam.ac.uk

**Data Availability Statement:** The raw data used in this meta-analysis is available in the repository https://osf.io/gz47h/?view_only=3805c47f6eb3460dbb3139e2001a5572. The full

## Abstract

### Background

Bisphosphonate drugs can be used to improve the outcomes of women with breast cancer. Whilst many meta-analyses have quantified their potential benefits for patients, attempts at comprehensive quantification of potential adverse effects have been limited. We undertook a meta-analysis with novel methodology to identify and quantify these adverse effects.

### Methods

We systematically reviewed randomised controlled trials in breast cancer where at least one of the treatments was a bisphosphonate (zoledronic acid, ibandronate, pamidronate, alendronate or clodronate). Neoadjuvant, adjuvant and metastatic settings were examined. Primary outcomes were adverse events of any type or severity (excluding death). We carried out pairwise and network meta-analyses to estimate the size of any adverse effects potentially related to bisphosphonates. In order to ascertain whether adverse effects differed by individual factors such as age, or interacted with other common adjuvant breast cancer treatments, we examined individual-level patient data for one large trial, AZURE.

### Findings

We identified 56 trials that reported adverse data, which included a total of 29,248 patients (18,301 receiving bisphosphonate drugs versus 10,947 not). 24 out of the 103 different adverse outcomes analysed showed a statistically and practically significant increase in patients receiving a bisphosphonate drug compared with those not (2 additional outcomes that appeared statistically significant came only from small studies with low event counts and no clinical suspicion so are likely artifacts). Most of these 24 are already clinically recognised: 'flu-like symptoms, fever, headache and chills; increased bone pain, arthralgia, myalgia, back pain; cardiac events, thromboembolic events; hypocalcaemia and osteonecrosis of the jaw; as well as possibly stiffness and nausea. Oral clodronate appeared to increase

analysis is available online at https://chjackson.github.io/adverse/vignettes/bisph_app2_index.html Individual patient-level data we were given access to by the trial owners and so we do not have the rights to share.

**Funding:** The authors received no specific funding for this work.

**Competing interests:** The authors have declared that no competing interests exist.

the risk of vomiting and diarrhoea (which may also be increased by other bisphosphonates), and there may be some hepatotoxicity. Four additional potential adverse effects emerged for bisphosphonate drugs in this analysis which have not classically be recognised: fatigue, neurosensory problems, hypertonia/muscle spasms and possibly dysgeusia. Several symptoms previously reported as potential side effects in the literature were not significantly increased in this analysis: constipation, insomnia, respiratory problems, oedema or thirst/ dry mouth. Individual patient-level data and subgroup analysis revealed little variation in side effects between women of different ages or menopausal status, those with metastatic versus non-metastatic cancer, or between women receiving different concurrent breast cancer therapies.

## Conclusions

This meta-analysis has produced estimates for the absolute frequencies of a range of side effects significantly associated with bisphosphonate drugs when used by breast cancer patients. These results show good agreement with previous literature on the subject but are the first systematic quantification of side effects and their severities. However, the analysis is limited by the availability and quality of data on adverse events, and the potential for bias introduced by a lack of standards for reporting of such events. We therefore present a table of adverse effects for bisphosphonates, identified and quantified to the best of our ability from a large number of trials, which we hope can be used to improve the communication of the potential harms of these drugs to patients and their healthcare providers.

## Introduction

Women with breast cancer face a choice of potential treatments and, in consultation with their healthcare professionals, need to consider the potential benefits and harms of each in order to reach an informed choice [1]. Considerable work has gone into quantifying the potential benefits of each treatment for different subgroups of patients (eg. [2–6]), and displaying those potential benefits in a clear way through websites such as Predict:Breast Cancer (breast.predict.nhs.uk), but rather remarkably the potential harms have not been quantified in the same way. We undertook this study as the result of consultation with breast cancer patients on what information they wanted when making treatment decisions.

To answer their requests, we aimed to determine, for women with breast cancer receiving bisphosphonates–

- What adverse events are they likely to experience, and with what frequency (in comparison with those not receiving bisphosphonates)?

- What is the likely severity of those adverse events?

- Do those events vary by drug or drug class, dose, means of delivery or other adjuvant therapy being received at the same time?

- Do those events vary by patient characteristics such as menopausal status or cancer stage?

Bisphosphonates drugs have been used since the 1990s including for the reduction of bone-related side effects such as pain due to secondary cancer and treatment-related osteoporosis

(particularly from aromatase inhibitors) [7,8]. More recently, studies have suggested that bis-phosphonates reduce the risk of breast cancer spreading to the bone [9,10] and improve sur-vival in post-menopausal women [9–11]. There are two classes of bisphosphonate: non-nitrogenous (which include the drug clodronate), and nitrogenous (which includes the drugs pamidronate, alendronate, ibandronate and zoledronate). The two have different modes of action and potentially different adverse effects [12].

Poon et al. conducted a meta-analysis of adverse events reported in cancer patients by 25 trials across a range of solid tumours, and involving either bisphosphonates or the RANKL inhibitor denosumab (used for the same purpose therapeutically) [13]. They identified 5 adverse effects reported significantly more frequently in patients taking a bone-modifying drug than in those taking placebo: vomiting, osteonecrosis of the jaw, combined nausea/vomit-ing, haematological and lymphatic toxicities, and respiratory problems. Domschke & Schuetz in 2014 [14] reported flu-like symptoms, nephrotoxicity, hypocalcaemia and osteonecrosis of the jaw. Other papers describe dose-related gastro-intestinal problems with oral nitrogenous bisphosphonates [15]. After the earliest reports of osteonecrosis of the jaw (ONJ) associated with bisphosphonate use in 2003 [16], most studies have taken steps to protect women taking bisphosphonates who need dental treatment, and kept close records of any symptoms of ONJ. Hypocalcaemia is also a well-recognised potential side-effect of bisphosphonate drugs.

The acute flu-like syndrome (known as the Acute Phase Response or APR) is associated particularly with the first infusion with intravenous zoledronic acid [17,18]. Reid et al. found a large number of symptoms in the 3 days following infusion, including 'flu-like symptoms, (fever, chills etc), gastrointestinal, eye inflammation, fatigue, dizziness, thirst and (more rarely) fainting, oedema, a cold, insomnia and tremors [19]. Additionally, some studies have identi-fied atrial fibrillation and stroke as potential longer term adverse effects of bisphosphonates [20,21], as well as renal dysfunction [22,23]. The British National Formulary lists 27 side effects with a frequency of over 1 in 100: alopecia; anaemia; arthralgia; asthenia; constipation; diar-rhoea; dizziness; dysphagia; electrolyte imbalance; eye inflammation; fever; gastritis; gastroin-testinal discomfort; headache; influenza like illness; malaise; myalgia; nausea; oesophageal ulcer; oesophagitis; pain; peripheral oedema; renal impairment; skin reactions; taste altered; vomiting [24].

We searched the literature for randomised trials involving any bisphosphonate drugs being tested in women with breast cancer and carried out a network meta-analysis of the adverse events data.

## Methods

### Data sourcing

*"What adverse events are women with breast cancer likely to experience?"*

For this, data was sought from randomised controlled trials in women with breast cancer where at least one arm received bisphosphonates and one arm did not and which reported adverse events.

We therefore conducted an initial search of the literature for relevant trials and reviews in PubMed, Cochrane Central Register of Controlled Trials, and Web of Knowledge databases using the search terms: Bisphosphonate AND (breast cancer) AND (toxicity OR adverse OR side effect OR safety OR efficacy). The search was done for all time periods up until June 2018, with no restriction on publication language. An update to the search was carried out up to December 2019, adding trials that had reported in that period.

Inclusion criteria were: only systematic reviews or randomised controlled trials where the population was women with breast cancer, the trial was randomised and had at least one arm containing a bisphosphonate drug.

Only trials that gave quantified data on at least one adverse event were included in the meta-analysis.

## Additional data sources

*"Do those events vary by other adjuvant therapy being received at the same time?"*

Because the meta-analysis data was too coarse to be able to answer this question, individual patient-level data was additionally sought from the large trial AZURE (NCT00072020, ISRCTN79831382) [25]. This trial, testing adjuvant zoledronic acid against observation only, involved 3,340 women across several countries and allowed women to receive chemotherapy, trastuzumab and/or hormone therapy in addition to bisphosphonates. The 'intense' dosing regime used in AZURE, although not one in common clinical use is ideal for investigating the presence of toxicity, especially synergistic interaction with chemotherapy [25], given the large size of the dataset, as long as it is not used on its own to determine absolute incidence frequencies. The congruence of the results from AZURE with other trials was also confirmed statistically.

## Data extraction

The adverse events data were extracted into a spreadsheet. Where adverse events were not reported, attempts were made by email to contact the study authors to request the data.

When necessary, patient numbers were back-calculated from reported percentages (rounding to the nearest whole number). Wherever possible, figures were used for the number of patients treated per protocol (e.g. the 'safety population') rather than the number randomised to it (intention to treat). Data were extracted from a trials registry website when possible. If results were not published in a trials registry, all published papers regarding a trial were examined for adverse data, and data was taken from the most recent paper that included adverse data (i.e. with the longest follow-up time possible). When the reported number completing the 'per protocol' trial in each arm varied between reports, the lowest number reported was used.

*"What is the likely severity of those adverse events?"*

To address this, details were recorded on the severity grade of each adverse event. When graded separately on a scale (0–5), the proportion of patients experiencing each grade was recorded. When 'serious' and 'non serious' events were reported (for example, as on clinicaltrials.gov), these were recorded as Grades 3/4 and Grades 1/2 respectively. Where adverse events were reported without indication of seriousness, these were recorded as 'ungraded'. Different terms likely to be used for the same symptom were combined (e.g. 'pyrexia' and 'fever'). 'Dizziness' and 'vertigo' were also combined as some trials reported them together. Full serious adverse event recording on clinicaltrials.gov often included single events. Only symptoms where more than 2 events were recorded in at least one trial were extracted. Some rarer events, such as many specific infections, cardiac symptoms (other than congestive heart failure) or non-breast cancer neoplasms, were collected together under a single heading. This means that for these events, if a single individual suffered 3 different kinds of infection, say, it would be recorded as 3 out of n experiencing 'infection'. Where 'osteonecrosis' was reported, it was assumed to be of the jaw. The number of withdrawals from the trial due to adverse events was also collected.

*"Do those events vary by patient characteristics such as menopausal status or cancer stage?"*

*"Do those events vary by drug or drug class, dose, means of delivery?"*

To address these questions, data were also collected on whether the therapy was adjuvant or neoadjuvant, the drug given, dose and mode of delivery, additional treatments the patients received, length of follow-up, whether the women were pre or post menopausal, and whether they had bone metastases.

Additionally, data were collected on aspects that allow a judgement of the quality of the evidence and risk of bias: funding source of trial, blinding, how fully the adverse events were reported and the number of patients in the 'intention to treat' and the 'per protocol' categories of the trial. These are reported in S1 Table and the raw dataset.

The database searching, abstract selection and data extraction was done independently by two reviewers (ZS and AF) and any differences resolved by referring back to the original source.

## Classifications of treatments for comparison

We started with the assumption that each different bisphosphonate drug, and each different dosing regime, might create a different profile of adverse effects. We then tested this assumption by comparing that model with other, simpler models. We did this for each symptom.

There is a different set of trials that reported data for each symptom, giving a series of symptom-specific networks of comparisons between treatments.

Because of the wide range of drugs, doses and delivery methods in the trials, five alternative classifications of treatments were considered. From the finest to the coarsest grouping, these are:

1. Different *doses and delivery methods* of different drugs considered as different treatments (See Fig 4 in S1 File).

2. Different *delivery methods* considered as different treatments, but different doses of the same drug considered as the same treatment.

3. Different *drugs* considered as different treatments, but different doses and delivery methods of the same drug considered as the same treatment.

4. *Nitrogenous* bisphosphonates and non-nitrogenous bisphosphonates (just clodronate) considered as different treatments.

5. *All bisphosphonates* considered as the same treatment.

Non-bisphosphonate control groups were distinguished as either placebo, observation only or denosumab. One study [26] had four arms, with two pairs of bisphosphonate and control comparisons distinguished by two different hormone therapies given, and this was treated as two different two-arm studies comparing bisphosphonate and control. A further four-arm study [27] had two bisphosphonate-control comparisons distinguished by different kinds of chemotherapy given, and this was treated in the same way.

## Risk of bias

The standard risk of bias tools are not designed for assessing bias in adverse events reporting. All trials collected adverse events as reported by clinicians rather than patients, and many stated that the adverse events recorded were only those considered by the clinician to be related to (or potentially related to) the treatment, which is likely to be common practice.

Rather than a formal risk of bias assessment we therefore present information to allow readers to make their own assessment of the data quality.

The biggest potential source for risk of bias were judged to be differences in recording and reporting of the adverse events. Some papers mentioned only a few non-systematically recorded adverse symptoms whilst others had much more full reporting. We therefore categorised papers on the basis of their reporting of adverse events: category 1 if the adverse event reporting threshold was whether an event occurred in $> = 5\%$ of patients, category 2 if it was between 5–10%, category 3 if fewer adverse events than this were reported or the threshold was unclear.

We also collected data on the key features of a risk of bias assessment: blinding (double blinded or open label), and the number of patients lost to follow-up. These are stated in S1 Table. Data on the funding source of the trial and on the length of follow-up (acute, during medication, long term), is reported in the raw dataset.

## Analysis

Each of the 103 recorded adverse effects was analysed as a separate outcome.

### Standard meta-analysis of the data in 56 trials: Bisphosphonate vs control

*"What adverse events are they likely to experience, and with what frequency (in comparison with those not receiving bisphosphonates)?"*

A standard fixed-effects meta-analysis was performed for each symptom independently, using studies that directly compared a bisphosphonate with a non-bisphosphonate control. The odds ratio and risk difference for the binary symptom outcome, compared between bisphosphonate and control, was estimated for each study. The Mantel-Haenszel method was then used, through the "meta" R package [28], to compute the pooled odds ratios and risk differences. Note that fixed-effects meta-analysis still gives a valid estimate of a weighted average of the study-specific effects, even when, as in this analysis, there is heterogeneity between the study-specific estimates [29].

*"Do those events vary by patient characteristics such as menopausal status or cancer stage?"*

The meta-analyses of the studies with direct comparisons of a bisphosphonate with a non-bisphosphonate control were extended to fixed effects meta-regression analyses [30]. The studies were categorised according to whether they included pre-menopausal, post-menopausal women or a mixture, and according to whether the participants had bone metastases (no participants, all participants or a mixture). The studies with a mixture of patient types were excluded, then the treatment effect (log odds ratio or risk difference) was modified with a linear regression term contrasting "all metastatic patients" or "all post-menopausal patients", with none.

### Network meta-analysis of the data in 56 trials: Different bisphosphonates compared

*"Do adverse events vary by drug or drug class, dose, means of delivery?"*

This question required the incorporation of both direct and indirect evidence, and so a network meta-analysis model was developed for each symptom independently. The following procedure was used to obtain a best-fitting network meta-analysis model that uses all available and relevant data, which requires a connected network of treatment comparisons.

For each potential side effect symptom, the available data consist of the subset of trials which report data on that symptom. The treatment networks were summarised graphically to aid appraisal of the evidence supporting each treatment effect (eg. Fig 2A in S1 File). If the network of treatment comparisons for this subset was disconnected under the most detailed model (grouping 1 in the 'Classifications of treatment for comparison' section above, which distinguishes doses and delivery methods), but became connected when placebo and observation-only controls were considered to be the same treatment, then these two control groups were considered to have identical risks of that symptom. Otherwise these controls were considered separately. Then, starting with the most detailed model (grouping 1), the treatments were grouped until the resulting treatment network was connected. Network meta-analysis models were fitted on treatment networks ranging from the finest connected network (1) to the most coarse (grouping 5 above, all bisphosphonates considered together). The coarser models make stronger assumptions about consistency of treatment effects, whereas the more detailed models may be driven by random variability in the data. Therefore, the preferred model was selected from among those which fitted successfully, using the deviance information criterion [31], a measure of optimal fit and predictive ability. If no network meta-analysis model could be fitted in practice due to sparsity of information, the results for that symptom were reported from standard meta-analyses of the studies reporting direct comparisons against a non-bisphosphonate control, for each alternative bisphosphonate treatment under the finest possible treatment classification. Further details of the network meta-analysis specification and implementation are given in the Supporting Information. Effects were deemed statistically and practically significant if they had an odds ratio with a posterior median of 1.5 or greater and a lower 95% credible limit greater than 1, from either the best-fitting network meta-analysis model or the fixed effects meta-analysis of studies that compared the corresponding treatment with observation only. Effects were also reported as significant if they had a posterior median difference in absolute risk of 0.02 or more between treated and control patients, and a credible limit greater than 0, for any definition of control.

To assess the sensitivity of our findings to the variations in levels of reporting of adverse events between trials, we repeated the analysis using only those trials that reported adverse events to the fullest extent (reporting all events that occurred with a frequency of <5% or higher).

## Analysis of absolute risks

In order to provide absolute risks of adverse events under each treatment (answering our question 'with what frequency?' women suffer these events), both an odds ratio and a baseline risk are required. Two potential baselines were sought. The first, for women undergoing breast cancer therapy, were taken as the pooled effects from standard meta-analysis of the direct data from the 'observation only' arms of the bisphosphonate trials used in this study, including only studies with the most complete level of reporting. These women were taking a variety of other anti-cancer therapies such as chemotherapy and hormone therapy and so are likely to have relatively high baseline levels of symptoms that could be caused by the cancer or other treatments they were undergoing.

For comparison, suitable absolute risks were sought from trials involving women of a similar demographic who were not suffering from cancer or taking any significant other medications. Searching clinicaltrials.gov for trials with reported results involving postmenopausal

women at a high risk of breast cancer and which had a placebo group led us to use data from trial NCT00083174 [32], a study of exemestane versus placebo for postmenopausal women at high risk of breast cancer (n = 2248 in the placebo group).

### Analysis of individual patient-level data in the AZURE trial

*"Do adverse event frequencies vary by patient characteristics or other adjuvant therapy being received at the same time?"*

Addressing this question required individual-level data, which was taken from a single large trial, AZURE. These data were used to estimate the overall risks of adverse events, and the relative risks of adverse events comparing individuals in the bisphosphonate and control groups, for various patient subgroups. Individual-level data also allowed investigation of whether some adverse effects co-occurred significantly more often with each other than might be expected at random, providing additional information to patients.

Women were sub-grouped by age, ER status of their tumour, HER2 status of their tumour and menopausal status. There was not enough variation in the population by ethnicity to carry out an analysis looking at the effect of ethnicity on adverse effects. In addition, we compared subgroups receiving different concomitant treatment regimes, categorised as endocrine therapy alone, chemotherapy alone, or both. Chemotherapy regimes were further categorised into anthracyclines only, taxanes only or both. This data also allowed consideration of whether any side effects co-occurred significantly, which would be important information for patients.

The events selected for this analysis included those for which there was a significant effect of bisphosphonates reported in the network meta-analysis, and those for which there was prior evidence in the literature of a link with bisphosphonates. In addition, we selected events for which the AZURE individual level data showed a statistically ($p<0.05$) and practically significant risk difference (estimate $>0.02$) or risk ratio (estimate $>1.5$).

## Results

The initial literature search identified 1207 titles and abstracts. Reviewing these produced 21 reviews and 98 published trials (for example, reported on clinicaltrials.gov) that appeared from the abstracts to fulfil our inclusion criteria. Additional references from the reviews produced 40 more trials.

280 full papers or reports were obtained for these 138 trials, and ineligible trials excluded, leaving 101 relevant trials (a list of these is available at DOI 10.17605/OSF.IO/GZ47H). Of these, 56 trials were identified which both met our inclusion criteria and had adverse events numerically reported for the individual arms (see S1 Table). These trials covered a total population of 18,301 patients receiving bisphosphonate drugs, versus 10,947 receiving no bisphosphonate (either an alternative such as denosumab, or a placebo, or observation only) from at least 46 different countries.

Fig 1 indicates the flow chart of study inclusion.

The data extracted from the 56 trials that fulfilled our criteria is available in the Open Science Foundation repository shown at the end of this paper. S1 Table shows the study characteristics.

### Meta-analysis of trials

Fig 2 shows, from the commonest at the top to the rarest at the bottom, the raw frequencies of each of the 103 adverse events (across all arms) in the 56 trials (left margin), and the results of

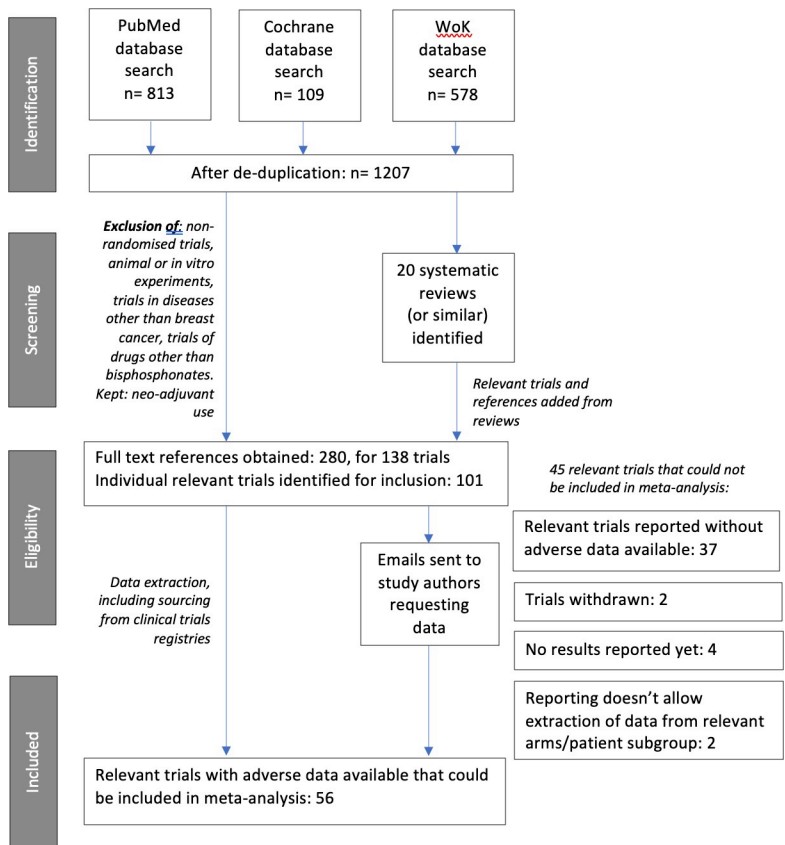

**Fig 1. PRISMA diagram showing data collection process.**

a fixed-effects meta-analysis, simply comparing the risks of suffering each adverse event in those taking bisphosphonate with the risk in those not taking bisphosphonates. Events where there is a large difference in risk (difference >0.02) or in odds ratio (>1.5) are highlighted black. There are 16 of these potentially significant adverse events: arthralgia, fatigue, nausea, increased bone pain, myalgia, back pain, nail changes, cough, abdominal pain, influenza-like symptoms, fever, hypocalcaemia, stiffness, hypertonia/muscle spasms, chills and osteonecrosis of the jaw. This simple analysis, however, does not take into account the different drug regimes and doses across these studies.

When the meta-regression results for all adverse events were filtered to include only those where the contrast in the treatment effect between subgroups was statistically significant, and the treatment effect on adverse event risk for the higher-risk subgroup was practically significant (odds ratio > 1.5 or risk difference > 2%), the only significant subgroup contrast when considering menopausal status is for insomnia (OR 0.22 (0.05, 0.95) in pre-menopausal patients and 1.81 (0.63, 5.24) in post-menopausal patients; Risk difference -0.16 (-0.31, -0.02) in pre-menopausal patients and 0.02 (-0.01, 0.05) in post-menopausal patients). However, this result is based on only 11 and 15 events in the two subgroups respectively and the estimated risk increase in post-menopausal women is only 2%.

Comparing non-metastatic and metastatic patients, there was a statistically and practically significant difference in adverse bisphosphonate effects between these groups for 9 adverse events (data and estimates shown in S2 Table). For fatigue and hypocalcaemia, the adverse effects were only significant for metastatic patients, among whom the odds ratio between treatment and

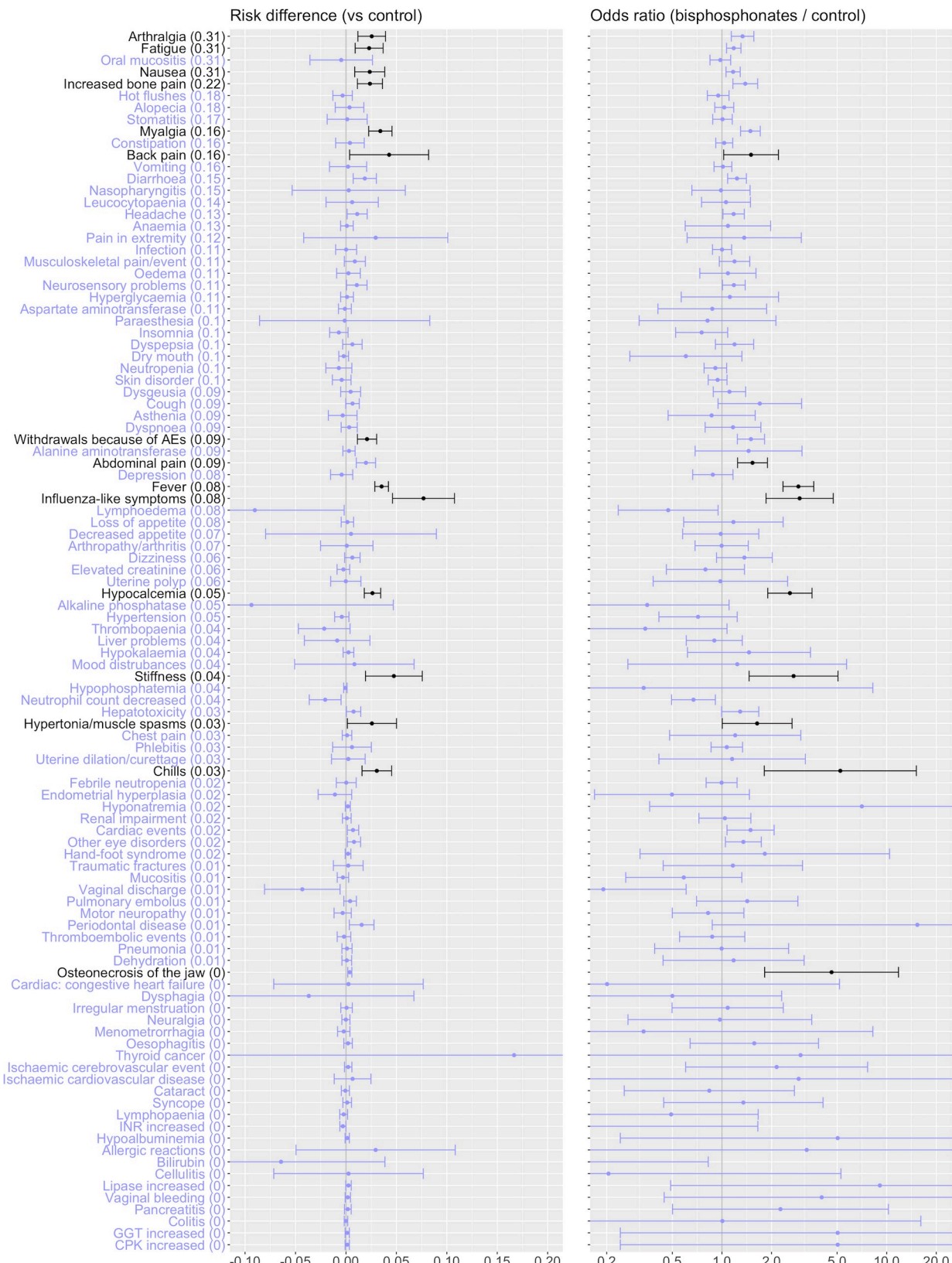

**Fig 2. Adverse events recorded in 56 randomised controlled trials testing bisphosphonates over no bisphosphonates.** The left panel is the estimated difference in risk (with 95% confidence intervals) between the bisphosphonate and no-bisphosphonate arms from a fixed effects meta-analysis of studies with this comparison, and the right panel is the estimated odds ratio. Highlighted black are those adverse events where the treatment effect is both statistically and practically significant, where practical significance is defined by an estimated odds ratio >1.5 or risk difference >0.02.

control was 1.69(1.24, 2.29) (risk difference 0.11(0.06, 0.17)) for fatigue and 2.14(1.19, 3.85) (risk difference 0.06(0.03, 0.08)) for hypocalcaemia. For increased bone pain, the effects were only significant for non-metastatic patients, with an odds ratio of 1.53(1.26, 1.85) (risk difference 0.02(0, 0.04)), though this might be due to differential reporting of this event between the groups. For dizziness, adverse effects were only significant in non-metastatic patients (odds ratio 2, though the risk difference was only 1%). For cardiac events, flu-like symptoms, neuralgia, and osteonecrosis of the jaw, only non-metastatic patients appeared to show significant effects of bisphosphonates, while for neutropenia, effects were only significant in metastatic patients, but these results are all based on small numbers of events (see S2 Table), so may not be reliable.

The meta-analysis hides considerable heterogeneity between individual studies. For example, taking the 24 arms of the 21 studies that reported rates of fever in patients given a bisphosphonate compared to a non-bisphosphonate control, the estimated baseline risks of the event vary widely (see S1 Fig). This is likely due to difference in the thresholds of recording or reporting symptoms: some only reported adverse events graded as 'serious'. It may also be due to differences between the patient populations and drug regimes in different trials (including other treatments that they are taking). However, the odds ratios and risk differences are more homogeneous between studies, indicating an increased risk of fever for people given bisphosphonates. This suggests that variable recording or reporting quality does not substantially influence the estimates of the treatment effects on risk.

## Network meta-analysis

For all 103 events for which there was evidence on the effects of bisphosphonates, the network meta-analysis results are illustrated in the same format in an online appendix (https://chjackson.github.io/adverse/vignettes/bisph_app2_index.html). We report in Table 1 those adverse effects for which a treatment effect was detected that was both practically and statistically significant.

Our sensitivity analysis using only the fullest-reporting trials (of which there were 24) confirmed the findings of most of the events listed in Table 1. In a few cases it did not. These cases are highlighted with as asterisk (*) in Table 1 and their colour made yellow rather than green. These results should therefore be treated with more caution. Two new events became significant in this subset of well-reported trials: neutropenia and skin disorders, but the certainty of these results should again be treated with caution. These effects are more usually associated with chemotherapy drugs.

In order to assist the judgement of the certainty of this evidence we have included columns in Table 1 to show the number of trials and number of patients whose data have been used to form both the direct and indirect odds ratio. Side effects whose odds ratios are based on direct evidence from a small number of trials and fewer than 1000 patients are also highlighted in yellow. Those with the lowest level of support are in orange.

In 15 out of the 157 treatment effects against observation-only from the best-fitting network meta-analysis models over all events, there was both direct and indirect evidence for the same treatment effect on the same event. In each case, the 95% credible limits for the direct vs indirect odds ratio spanned 1, indicating no important conflict between direct and indirect evidence and providing reassurance about assumptions of transitivity.

Table 1. Adverse effects of bisphosphonate drugs in women with breast cancer.

| Outcome | Treatment | Odds ratio from network meta-analysis | No. studies forming basis of network meta-analysis odds ratio (total n) | Odds ratio from direct comparisons | Direct comparison studies (total n) | Number of women (out of 100) who would be expected to experience the outcome. Taking treatment: | | 'serious' or Grade 3+ (absolute figures) |
|---|---|---|---|---|---|---|---|---|
| | | | | | | NO | YES | |
| **'Influenza-like symptoms'** * | Any bisphosphonate | 16.09 (2.04,207.85) | 12 (9,968) | 22.18 (2.98, 164.89) | 7 (1,142) | 3 | 37 (7–88) | 5% (35/718) |
| **Fever** | Zoledronic acid | 5.2 (2.88,8.79) | 31 (25,388) | 4.06 (3.06–5.38) | 18 (10,581) | 1 | 6 (4–10) | 9% (82/933) |
| | Pamidronate | 4.87 (1.9–12.62) | 31 (25,388) | 1.52 (0.47–4.92) | 2 (1,047) | 1 | 6 (2–14) | 100% (3/3) |
| **Chills** | Zoledronic acid (no direct data for other drugs) | 15.19 (3.96,94.86) | 8 (10,368) | 8.55 (2.26, 32.32) | 7 (4,581) | 2 | 26 (8–68) | 1% (3/213) |
| **Headache** | Zoledronic acid | 1.48 (1.06, 2.25) | 28 (24,733) | 1.19 (1.02, 1.4) | 15 (9,354) | 24 | 32 (25–41) | 3% (36/1304) |
| **Arthralgia/joint pain** | Nitrogenous bisphosphonate | 1.54 (1.28, 1.86) | 31 (21,683) | 1.53 (1.29, 1.81) | 21 (7,376) | 17 | 24 (20–27) | 4% (189/5071) |
| **Myalgia** | Zoledronic acid | 2.36 (1.61, 3.6) | 23 (22,626) | 1.52 (1.31, 1.75) | 14 (8,158) | 3 | 8 (5–11) | 3% (47/1343) |
| * | Pamidronate | 2.32 (1.09, 5.21) | 23 (22,626) | | | 3 | 8 (4–16) | |
| * | Oral ibandronate | 2.04 (1.13, 3.85) | 23 (22,626) | | 1 (620) | 3 | 7 (4–12) | 6% (39/611) |
| **Stiffness** * | Zoledronic acid (data not available for other drugs) | 2.76 (1.27, 5.83) | 3 (1,873) | 2.72 (1.46, 5.06) | 2 (971) | 2 | 4 (2–8) | 0% (0/37) |
| **Hypertonia/muscle spasms** | Zoledronic acid | 1.52 (0.93, 2.81) | 8 (4,613) | 1.53 (1.02, 2.3) | 7 (4,579) | 4 | 6 (4–11) | 14% (15/106) |
| **Back pain** | Nitrogenous bisphosphonate | 1.64 (1.08, 2.69) | 15 (12,076) | 1.54 (1, 2.37) | 9 (3,652) | 8 | 13 (9–19) | 3% (37/1247) |
| **Increased bone pain** in all patients | Any bisphosphonate | 1.79 (1.33, 2.49) | 31 (21,995) | 1.73 (1.47, 2.04) | 20 (10,646) | 11 | 19 (15–24) | 10% (352/3386) |
| **Increased bone pain** in patients with non-metastatic cancer | Any bisphosphonate | | | 1.53 (1.26,1.85) | 11 (5,984) | 11 | 16 (14–19) | |
| **Hypocalcaemia** in all patients | Zoledronic acid | 7.28 (1.36,48.93) | 14 (16,853) | 7.07 (0.87, 57.56) | 5 (4,489) | 4 | 24 (6–68) | 3% (11/336) |
| | Pamidronate | 3.18 (1.2, 8.64) | 14 (16,853) | 3.23 (1.45, 7.22) | 1 (294) | 4 | 12 (5–27) | |
| | Ibandronate | 6.15 (1.16,41.76) | 14 (16,853) | | 2 (1,742) | 4 | 21 (5–64) | 2% (5/210) |

(Continued)

**Table 1.** (Continued)

| Outcome | Treatment | Odds ratio from network meta-analysis | No. studies forming basis of network meta-analysis odds ratio (total n) | Odds ratio from direct comparisons | Direct comparison studies (total n) | Number of women (out of 100) who would be expected to experience the outcome. Taking treatment: | | 'serious' or Grade 3+ (absolute figures) |
|---|---|---|---|---|---|---|---|---|
| | | | | | | NO | YES | |
| **Hypocalcaemia** in patients with metastatic cancer | Any bisphosphonate | | | 2.14 (1.19,3.85) | 8 (2,689) | 4 | 9 (5–14) | |
| **Cardiac events** (excluding congestive heart failure) | Nitrogenous bisphosphonate | 2.73 (1.33, 6.31) | 18 (20,236) | 2.51 (1.52, 4.17) | 11 (8,559) | 3 | 7 (4–15) | 62% (143/231) |
| | Non-nitrogenous bisphosphonate (clodronate) | 4.35 (1.44, 13.6) | 18 (20,236) | 5.22 (0.25, 109.67) | 2 (3,517) | 3 | 11 (4–27) | 32% (21/65) |
| **Osteonecrosis of the jaw** * | Nitrogenous bisphosphonate | | | 7.69 (2.31, 25.55) | 27 (26,018) 102 cases | | | |
| **Periodontal disease** * | Zoledronic acid (other drugs, with more evidence, not significant) | 10.2 (1.14,1322.28) | 2 (1803) only 16 events | 15.34 (0.87, 269.36) | 1 (903) | | | 14% (1/7) |
| **Nausea** * | Any bisphosphonate | 1.33 (1.11, 1.63) | 38 (25,049) | 1.21 (1.08, 1.35) | 27 (13,928) | 34 | 41 (37–46) | 9% (323/3579) |
| **Abdominal pain** | Any bisphosphonate | 1.51 (1.1, 2.11) | 24 (19,808) | 1.51 (1.13, 2.02) | 17 (10,165) | 13 | 19 (14–24) | 10% (122/1277) |
| **Diarrhoea** | Oral clodronate (also possibly true for others taken orally) | 1.81 (1.35, 2.74) | 27 (23,194) | 3.34 (1.18, 9.45) | 4 (4,759) | 11 | 18 (14–25) | 8% (49/591) |
| **Vomiting** | Clodronate | 1.67 (1.18, 2.92) | 22 (19,059) | | 2 (1,242) | 19 | 28 (22–41) | 3% (9/312) |
| **Hepatotoxicity** * | Oral ibandronate | 3.39 (1.13,11.73) | 10 (11,040) | 4.27 (1.28, 14.22) | 1 (2,800) | 26 | 55 (29–81) | 0% |
| | Clodronate | 1.46 (0.78–2.59) | 10 (11,040) | 1.69 (1.05–2.71) | 3 (4,586) | 26 | 34 (22–48) | 15% (14/95) |
| **Dysgeusia** | Zoledronic acid (evidence suggestive for others too) | 8.53 (1.25,119.8) | 4 (8,354) | 10.1 (1.17, 87.24) | 1 (301) | 0 | 1 (0–11) | |
| **Neurosensory problems** | Any bisphosphonate | 1.29 (1.05, 1.66) | 17 (19,922) | 1.23 (1.05, 1.45) | 13 (12,821) | 10 | 12 (10–15) | 3% (52/1509) |
| **Thromboembolic events** | Zoledronic acid | 2.74 (0.98, 7.65) | 9 (14,865) | 3.18 (1.34, 7.51) | | 2 | 5(2–14) | 72% (28/39) |
| | Ibandronate | 13.33 (2.43,73.09) | 2 (5,971) | | | 2 | 22 (5–60) | 91% (10/11) |
| * | Clodronate | 7.27 (1.41,33.44) | 12 (14,86) | | 1 (3,235) | 2 | 13 (3–40) | 100% (27/27) |

(Continued)

**Table 1.** (Continued)

| Outcome | Treatment | Odds ratio from network meta-analysis | No. studies forming basis of network meta-analysis odds ratio (total n) | Odds ratio from direct comparisons | Direct comparison studies (total n) | Number of women (out of 100) who would be expected to experience the outcome. Taking treatment: | | 'serious' or Grade 3+ (absolute figures) |
|---|---|---|---|---|---|---|---|---|
| | | | | | | NO | YES | |
| **Fatigue** in all patients | Any bisphosphonate | 1.28 (1.06, 1.65) | 34 (25,914) | 1.13 (1.02, 1.26) | 23 (14,490) | **10** | **13** (11–16) | 5% (247/4607) |
| **Fatigue** in patients with metastatic cancer | Any bisphosphonate | | | 1.69 (1.24,2.29) | 5 (1,556) | **10** | **16** (12–21) | |
| **Infection** * | Ibandronate (oral) | | 22 (23,314) | 2.24 (1.18, 4.22) | | | | |
| Additional findings from analysis of only 24 trials with the most comprehensive reporting of adverse events | | | | | | | | |
| **Neutropaenia** | Nitrogenous bisphosphonate | 2.33 (1, 5.43) | | 2.04 (0.7, 5.93) | | **4** | **9** (4–19) | 5% (8/166) |
| **Skin disorder** | Any bisphosphonate | 1.48 (0.87, 2.44) | | 1.61 (1.03, 2.48) | | **5** | **7** (4–11) | 3% (37/1142) |

Adverse events (and treatments) for which either the direct comparison or the selected network meta-analysis model produced a practically and statistically significant treatment effect, defined as an odds ratio with a posterior median of 1.5 or greater and lower 95% credible limit greater than 1, or a risk difference with posterior median 0.02 or greater and a positive lower 95% credible limit. The odds ratio is the ratio of the odds of the event between the treatment group (as defined in the table) and an "observation only" control group.

* Finding not upheld when analysis run on only 24 trials with comprehensive adverse event reporting (reporting threshold <5% of patients suffering the event).

Colours represent subjective quality of evidence (green = highest, yellow = lower, red = lowest, based on replicability of results between full dataset and comprehensive reporting subset, subgroup analyses, and number of patients included in direct comparison trials).

We report the proportion of each outcome that were rated as Grade 3+ or Serious in the trials. However, this is likely to be subject to high risk of bias as some trials only reported symptoms of these higher grades.

## Absolute risks

The absolute risks calculated on the baseline of 'women of a similar demographic without cancer' are reported in S3 Table. Many relevant outcomes were not recorded in these women–possibly because they did not occur, or because the trial investigators did not consider them relevant to the drug they were testing (exemestane). The absolute risks reported in Table 1 are from women with breast cancer in the 'observation only' arms of the trials in the meta-analysis.

Out of those where we have baseline data for both women without cancer and those undergoing cancer treatment, two (arthralgia/joint pain, and back pain) show close agreement in rates, whilst the other four (fatigue, cardiac events, nausea and diarrhoea) appear to be higher in the women undergoing cancer therapy and are likely side-effects of the cancer or other therapies.

Since the network meta-analysis allowed differentiation different drugs, drug classes, doses and means of delivery, where the network that best fitted the data differentiated on treatment characteristics, different odds ratios for different treatments are shown in Table 1.

## Analysis of individual patient-level data (for drug interactions and demographic differences)

Patients in the AZURE trial were mixed in terms of menopausal status (1488 premenopausal, 480 < = 5 years since menopause, 1011 >5 years since menopause, 316 no data) with an age range of 20–87 (mean 51), were mostly ER positive (2581 ER +ve, 695 ER -ve, 19 no data), mostly HER2 negative (1230 HER2 -ve. 408 HER2 +ve, others unknown), and mostly Caucasian (3121 Caucasian, 149 other, 25 no data).

First, the overall data was analysed to estimate the percentage risk difference between those given bisphosphonates and those not given bisphosphonates, and the risk ratio, for each adverse event reported.

After selecting adverse events for which there was already a suggestion of a link with bisphosphonates, the only additional event that showed a significant effect in the AZURE data was arthritis (a fact which illustrates the congruency of the AZURE data with those from other trials).

Within this list, some categories of events were grouped together (myalgia with arthralgia, fever with chills and flu-like symptoms, jaw with dental problems, gastrointestinal issues, mucositis and stomatitis, and cardiac symptoms). For four events there was not enough data to do sub-group analysis (DLCO, thrombocytosis, menorrhagia, carpal tunnel syndrome). This left 15 adverse event categories for which we could carry out a subgroup analysis.

Event rates were not affected by age, ER status, HER2 status and menopausal status, apart from a possible increase in arthritis as a side effect reported for women under 50 compared to those over 50. Fever was also possibly more likely in women with ER positive tumours–which may be related to other adjuvant treatments they were taking relating to their tumour type (hormone therapy).

The data on the risk differences in these different subgroups is visualised in Fig 3.

To investigate whether adverse events varied by other concurrent therapies, the same data was analysed according to the treatment regimes that the patients were undergoing. 137 patients were taking endocrine therapy alone, 708 chemotherapy alone and 2450 both. This revealed only one potentially significant interaction: that women not receiving chemotherapy were recorded as suffering slightly more fever and headache. See Fig 4.

Within the category of women taking chemotherapy, we also investigated whether there were interactions between the side effects suffered from the bisphosphonates and the type of chemotherapy they were taking (2328 were taking anthracyclines only, 757 both anthracyclines and taxanes, 7 taxanes only, 203 neither). This yielded no significant interactions.

Correlations between the different adverse effects (see Fig 5) showed that leucopenia, neutropenia, anaemia commonly co-occurred, along with (to a lesser extent) thrombocytopenia, abnormal liver function tests (LFTs), hyperglycaemia and hypocalcaemia. A separate group of events that commonly co-occurred were alopecia, fatigue/lethargy, gastrointestinal events and mucositis/stomatitis. Myalgia/arthralgia, fatigue/lethargy and hot flushes also commonly co-occurred. Of these side effects, only fatigue and arthralgia/myalgia are associated with bisphosphonates in our meta-analysis. The other clustered symptoms seem likely to be effects caused by other drugs taken concurrently (most likely chemotherapy and hormone therapy).

## Discussion

In this study we have set out for the first time, to our knowledge, to produce a comprehensive list of the adverse effects of bisphosphonate drugs when used as part of the therapeutic regime in women with breast cancer. We also aimed to take into account the severities of those events and factors that might affect them, both related to the individual patient and the treatment regime. By using individual patient-level data as well as both a standard meta-analysis and a

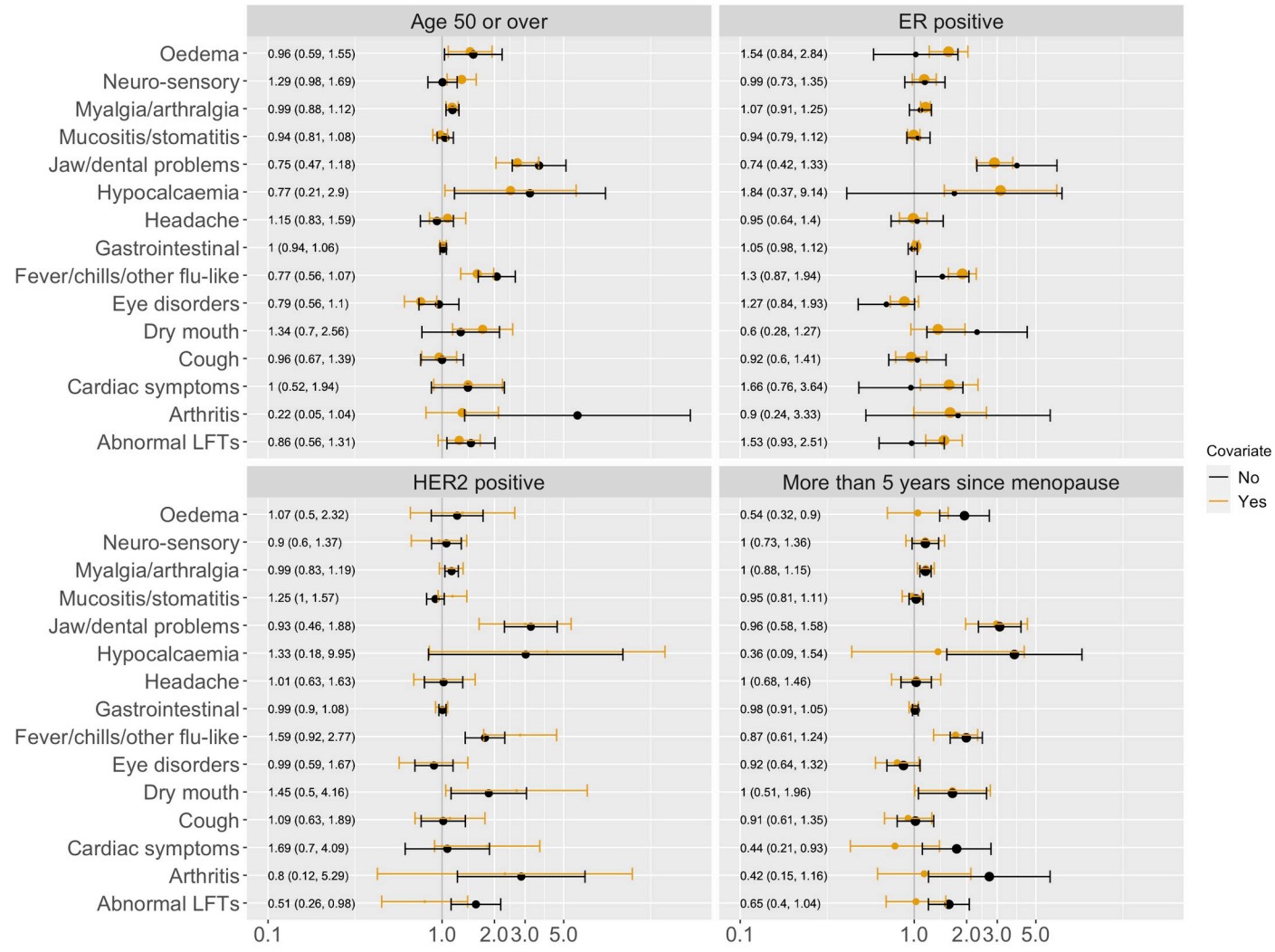

**Fig 3. Subgroup analyses of the individual patient data from AZURE.** Shows: The relative risk of each side-effect (bisphosphonates versus control) if they were in the indicated subgroup (yellow) or not (black). Separation between the yellow and black intervals indicates how much the covariate modifies the treatment effect.

network meta-analysis of all the data from trials involving bisphosphonates in women with breast cancer we hope to have obtained the best estimates currently possible of the odds ratios of adverse effects of different bisphosphonate drugs and regimes in order to be able to give advice to patients (and their clinicians) considering this therapy. We have sought appropriate baseline risks to combine with these relative effects, in order to provide a set of absolute risks which can be cited as simple frequencies in patient information. It should be noted, however, that these figures have relatively large uncertainties around them–both quantified and unquantified (in terms of the quality of the evidence they are based on which shows considerable heterogeneity). Additionally, we have extracted information on the likely severity of the symptoms. However, we acknowledge the substantial heterogeneity across populations and protocols and significant biases in reporting that would effect severity estimates so all our quantifications should only be considered as 'ballpark' figures.

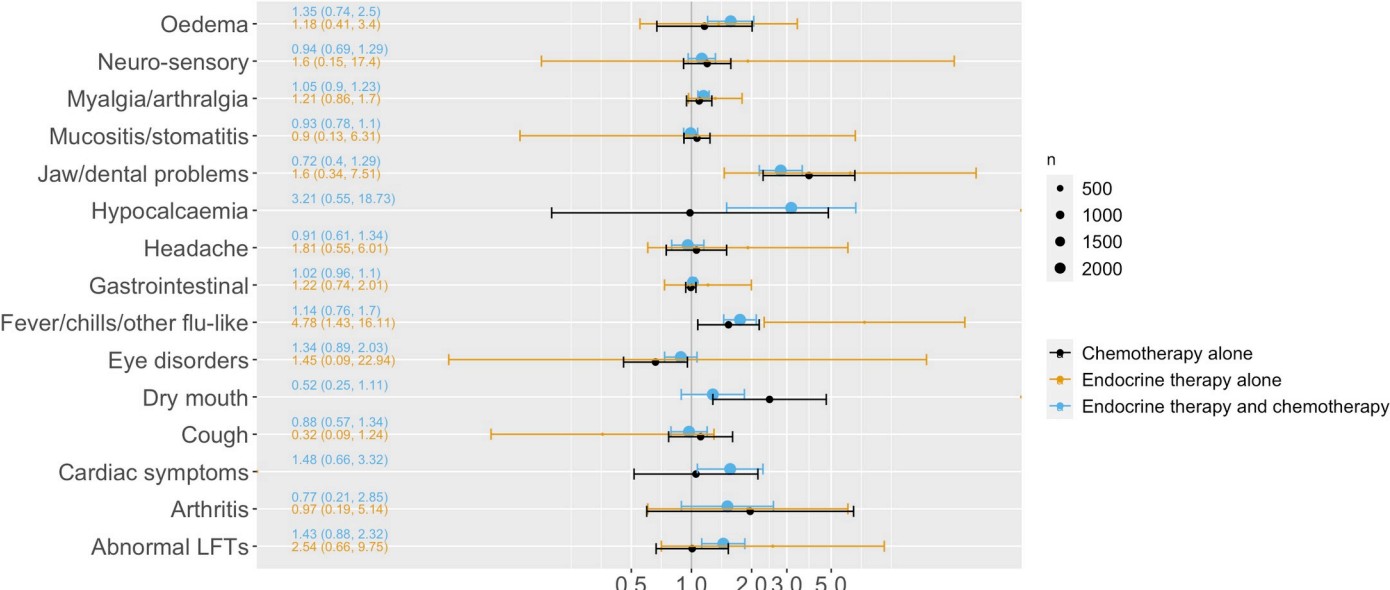

**Fig 4. Analysis of individual patient data in AZURE by concurrent treatment regime.** The relative risk of each side-effect for patients on different treatment regimes is shown alongside bisphosphonates.

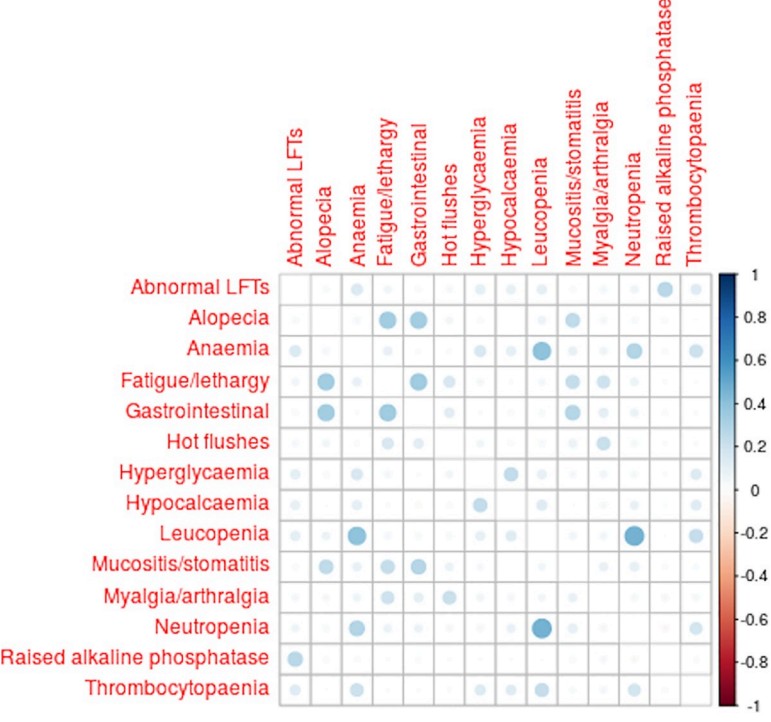

**Fig 5. Correlation coefficients between different adverse events, as observed in the patients in the AZURE trial randomised to zoledronic acid.** Area of dot is proportional to the correlation coefficient. The graph includes only the events which occurred in more than 5 out of these 1665 patients, and had a correlation of 0.2 or more with at least one other event. (produced using the corrplot R package [33]).

Answering our aim of investigating individual traits and concurrent drugs that might effect adverse effect profiles, the subgroup analysis and analysis of potential interactions between bisphosphonates and other, common concurrent therapies for breast cancer patients using individual patient-level data from the AZURE trial, suggests that the relative effects we have estimated are likely to be broadly applicable to all patients regardless of whether their cancer is metastatic or not and other concurrent cancer therapies with the exception of hypocalcaemia and fatigue (significant in patients with metastatic cancer only) and increased bone pain (significant in patients with non-metastatic cancer only, although this may be a reporting bias). Younger patients potentially suffer slightly disproportionately more joint pain (although this may also be a reporting bias). The fact that we found no evidence of interaction with other concurrent therapies (as is also the case with the benefits of bisphosphonates) means that the potential side effects can be considered additive to the side effects of other treatments being taken, raising the prospect of being able to model the individual side effect profile for a patient relative easily, in the same way that algorithms such as that behind the online tool Predict currently model the potential benefits.

The side effects for which this analysis has found an increased risk due to bisphosphonates when taken as part of a treatment regime for breast cancer are broadly in line with previous literature. Hypocalcaemia is increased from a risk of about 4 in 100 to around 20–25 out of 100, significant only in patients with metastatic cancer. In addition, around a third of patients are likely to suffer fever & chills (often grouped together as 'flu-like symptoms). As has been noted clinically, intravenous bisphosphonates are more likely to cause the 'flu-like response than those taken orally.

Possibly related symptoms that also showed a significant increase in patients taking bisphosphonates include headaches (risk raised from 24 out of 100 patients suffering to around 32 out of 100), arthralgia (risk raised from 2 out of 100 to around 3 out of 100), stiffness (raised from 17 out of 100 to around 24 out of 100), myalgia (raised from 3 out of 100 to around 8 out of 100), hypertonia/muscle spasms (raised from 4 out of 100 to around 6 out of 100) and back pain (raised from 8 out of 100 to around 13 out of 100). Increased bone pain, at least in patients without metastases, (found in 11 out of 100 patients not taking bisphosphonates but about 19 out of 100 taking bisphosphonates) is also found to be significant.

Abdominal pain (risk raised from 13 out of 100 to around 19 out of 100), diarrhoea (raised from 11 out of 100 to around 18 out of 100), vomiting (from 19 out of 100 to 28 out of 100) and possibly nausea (from 34 out of 100 to around 41 out of 100) were also increased risks, at least with oral clodronate. Some of these increased risks of gastrointestinal issues at least were true in trials of intravenous zoledronic acid as well, so appear not to be related only to some oral bisphosphonates.

Fatigue emerged as a potentially raised risk from bisphosphonates (more certain in patients with metastatic cancer), raising the risk from 10 in 100 to 13 in 100.

Hepatotoxicity is also a potential effect of oral bisphosphonates at least (risk raised from 26 out of 100 to 34–55 out of 100), and dysgeusia of zoledronic acid at least (risk raised from less than 1 out of 100 to between 0–11 out of 100) although this is weak evidence, based on only a few events.

Rarer side-effects include cardiac symptoms excluding congestive heart failure (risk raised from 3 out of 100 for these cancer patients to between 7–11 out of 100), thromboembolic events (risk raised from 2 out of 100 to 5–22 out of 100) and osteonecrosis (0 reported cases in those not taking bisphosphonates, 84 cases out of 18,301 taking bisphosphonates, although we were not able to distinguish different bisphosphonate drug regimes and calculate odds ratios), which should be guarded against and monitored, and neurosensory problems also emerged as a significant but only slightly increased side-effect (risk raised from 10 out of 100 to about 12

out of 100) based on heterogenous data–raised in two studies only. A significantly increased risk of <u>infection</u> emerged only from one trial (of oral ibandronate).

The fact that our figures do not conflict with clinical expectation and are in line with previous findings is reassuring. What makes this study particularly useful is that we can now put some numbers on the likelihoods of patients suffering each of these symptoms, and also their severity (see Table 1 for details).

We found no evidence of significant haematological and lymphatic toxicities or respiratory problems in line with Poon et al. [13] (although they were including the non-bisphosphonate drug denosumab in their study). We also did not find evidence of dizziness, thirst, fainting, eye disorders or insomnia as adverse events, as reported by Reid et al. [17] nor of renal dysfunction as reported by Tanvetyanon & Stiff [22] & Wilkinson et al. [20], although the meta-analysis of the trials in women without breast cancer did highlight oedema and thirst (as well as loss of appetite). Compared with the listing in the British National Formulary [24], we also saw no evidence of increased risks of alopecia, constipation, oesophagitis, and only limited evidence for skin reactions.

It is likely that some of the more widely-recognised risks, such as osteonecrosis of the jaw and renal dysfunction, are now greatly reduced in trials (and the clinical population) by preventative measures such as dose adjustments and screening for exclusion factors.

Our study, however, does suffer from a number of limitations that make our conclusions less certain than might be hoped.

The first of these is the availability of good quality data on adverse effects. It is notable how much difference there has been since the introduction of the registry clinicaltrials.gov and Section 801 of the Food and Drug Administration Amendments Act (2007) which mandates the publication of the results–including adverse effects—of certain trials. Academic publications rarely include as much detail as the registry or Clinical Study Reports, which are also increasingly being made publicly available. This increase in the reporting, however, needs to be matched with an increase in the quality of recording of symptoms. All the symptoms we have analysed have been physician-recorded rather than patient-recorded–something that is known to give a considerable underestimation of the prevalence (see review in [34]). In addition, almost all trials overtly state that they record only adverse symptoms which 'the physician deems attributable to the treatment'. This inevitably adds substantial reporting bias: symptoms not recognised as related to a particular drug will not be recorded–and hence not recognised. There is also likely to be heterogeneity between trials in how adverse events were monitored and reported. On top of this, it is well-recognised that many patients do not take drugs exactly as per protocol, either unintentionally or intentionally–the latter may be particularly true if the side effects become greater, meaning that there may be further under-reporting as patients moderate their doses (and don't report it) in response to side effects [35]. Finally, the use of different terms and scales for recording and reporting adverse events makes it more challenging to concatenate them for a meta-analysis. These are well-recognised problems, particularly in older trials.

We are thus faced with considerable heterogeneity in the reported absolute risks from each trial. The relative risks are also heterogeneous but more consistent between studies, giving us hope of calculating odds ratios with a slightly higher level of certainty for those effects which are recorded.

We also lack enough information on the severity and intensity of side effects to be able to answer patients' questions on this. The data suffers from too great a recording and reporting bias to be able to ascertain reliable estimates of the proportions of patients who suffered more severe symptoms of each adverse event.

Having clear, reliable information on the likely frequency and severity of adverse effects is critical to patients (and clinicians) in order to allow them to make evidence-informed decisions about treatment options. Importantly, it also allows them to select prophylactic treatments and to be prepared for possible side-effects, potentially decreasing anxiety and helping them stick to their treatment plans [36,37].

Until now, the majority of meta-analyses have concentrated on the potential benefits and many have completely ignored the potential harms (with a few honourable exceptions). We hope that this study will inspire others to focus on improving the reporting and quantification of adverse effects. We are making the raw data from our literature extraction available to allow it to be easily reanalysed and updated, but hope that those with access to large amounts of individual patient-level data will publish analyses of their adverse events data, which would be a powerful addition to the literature. Future useful work would be mapping the timings of the adverse events, particularly giving patients a likely duration for the acute phase response, building on the work of Reid et al. [17].

In the meantime, we hope that Table 1 will be a useful guide for those advising patients considering bisphosphonates as part of their therapy for breast cancer, and we are carrying out further work on how best to present these results graphically for inclusion into risk communication tools such as Predict.

## Supporting information

**S1 Checklist.**
(DOC)

**S1 Fig. Meta-analysis of trials which directly compare the risk of fever between a group of patients receiving bisphosphonates and a non-bisphosphonate control group.**
Figure showing the trial-specific treatment effects on the event of fever as odds ratios and risk differences, alongside the baseline risks (proportion of the control group who have the event–right hand column). There is wide variability in the reported baseline risks on the right. However, the odds ratios and risk differences are more homogeneous between studies, suggesting that variable recording or reporting quality does not substantially influence the estimates of the treatment effects on risk. The treatment effect on risk is estimated as an odds ratio or risk difference for each study, with an estimate of baseline risk shown in the right panel, and pooled estimates at the bottom. Point estimates and 95% confidence intervals are given–note that some of the upper and lower limits exceed the axis range. In OPTIMIZE-2 there are two estimates, one for each of two bisphosphonate arms. The two pairs of arms in the ABCSG12 trials are shown on separate lines.
(PDF)

**S1 Table. Trials included in the analysis and their characteristics.**
(DOCX)

**S2 Table. Results of a fixed effects meta-regression analysis showing adverse effects where the contrast in the treatment effect between results in patients with or without metastatic breast cancer was statistically significant, and the treatment effect on adverse event risk for the higher-risk subgroup was practically significant (odds ratio > 1.5 or risk difference > 2%).**
(DOCX)

**S3 Table. Calculated absolute risks for some adverse effects (and treatments) where baseline risks were available for women of a similar demographic but without breast cancer**

**(taken from trial NCT00083174).**
(DOCX)

**S1 File.**
(DOCX)

## Acknowledgments

We are extremely grateful to the Clinical Trials Research Unit at the University of Leeds for providing the individual patient-level data for the AZURE trial.

We would like to thank Leila Finikarides for her help with the clinical terminology, several colleagues who reviewed and helped improve the manuscript, and all the trial authors who kindly responded to our queries and requests to clarify or expand on their adverse event data.

## Author Contributions

**Conceptualization:** Alexandra L. J. Freeman, David J. Spiegelhalter.

**Data curation:** Alexandra L. J. Freeman, Zśofia Szlamka.

**Formal analysis:** Christopher Jackson, David J. Spiegelhalter.

**Methodology:** Christopher Jackson.

**Project administration:** Alexandra L. J. Freeman.

**Supervision:** Alexandra L. J. Freeman.

**Writing – original draft:** Christopher Jackson, Alexandra L. J. Freeman.

**Writing – review & editing:** Christopher Jackson, Alexandra L. J. Freeman, Zśofia Szlamka, David J. Spiegelhalter.

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
