## [Decision Letter · Decision Letter 0]

25 Sep 2020

PONE-D-20-18874

The adverse effects of bisphosphonates in breast cancer: a systematic review and network meta-analysis

PLOS ONE

Dear Dr. Freeman,

Thank you for submitting your manuscript to PLOS ONE. After careful consideration, we feel that it has merit but does not fully meet PLOS ONE’s publication criteria as it currently stands. Therefore, we invite you to submit a revised version of the manuscript that addresses the points raised during the review process.

The reviewers that evaluated your manuscript have identified some strong issues with it, to which I invite you to have a look and see if you can respod appropriately. It is however probable that the revised submission will be re-assessed anew in light of your revisions.

We look forward to receiving your revised manuscript.

Kind regards,

Spyridon N. Papageorgiou, DDS, Dr Med Dent

Academic Editor

PLOS ONE

Journal Requirements:

2. Please include captions for your Supporting Information files at the end of your manuscript, and update any in-text citations to match accordingly. Please see our Supporting Information guidelines for more information: http://journals.plos.org/plosone/s/supporting-information

Reviewers' comments:

Reviewer's Responses to Questions

**Comments to the Author**

1. Is the manuscript technically sound, and do the data support the conclusions?

Reviewer #1: No

Reviewer #2: Partly

Reviewer #3: Yes

Reviewer #4: No

2. Has the statistical analysis been performed appropriately and rigorously? 

Reviewer #1: No

Reviewer #2: No

Reviewer #3: Yes

Reviewer #4: No

3. Have the authors made all data underlying the findings in their manuscript fully available?

Reviewer #1: No

Reviewer #2: Yes

Reviewer #3: Yes

Reviewer #4: Yes

4. Is the manuscript presented in an intelligible fashion and written in standard English?

Reviewer #1: Yes

Reviewer #2: No

Reviewer #3: Yes

Reviewer #4: Yes

5. Review Comments to the Author

Reviewer #1: The authors undertook a meta-analysis in order to describe more comprehensively than until now available the side effect profile of bisphosphonates. While this is a subject worth investigating, both because of the frequent use of bisphosphonates in many established clinical indications and the lack of systematic data, the attempt is severely hampered by a number of factors:

(1) most important: As the authors admit themselves, “…the analysis is limited by the availability and quality of data on adverse events, and the potential for bias introduced by a lack of standards for reporting of such events...“. I am afraid this essentially precludes a meaningful analysis, because the “garbage in – garbage out” principle cannot be “healed”…

Some of this is represented in the wide range of side effect frequencies reported (e.g. p3) – it does not make sense to the clinician to tell him/her that e.g. flu-like symptoms are increased to “….7-88%....”

(2) Clinically, it is more or less meaningless to analyze drug side effects in trials from setting as different as neoadjuvant, adjuvant, and metastatic. This structural error of the approach is another reason for the wide variation of results. Furthermore, this mixing everything together annihilates possibly meaningful results for the individual settings. E.g. “Fever” may be quite negligible for the metastatic patient but less than trivial for the one-time user in a neoadjuvant setting. Another example: Myalgia and arthralgia, which are anyway frequent side effects of many adjuvant drugs in breast cancer, may not play a “bisphosphonate-specific” role in that setting, but be extremely cumbersome in otherwise healthy osteoporosis of prostate cancer patients, or patients with triple-negative breast cancer who do not receive aromatase inhibitors.

(3) The lack of individual patient data reduces the value of the meta-analysis approach – having them available from just one trial (AZURE) that somewhat fall out of the average range both by its patient selection, excessive dose, and negative results heavily limits the meaningfulness of the results described.

(4) As a result of mixing together many different clinical settings, indications, and stages, concomitant medication of patients must significantly impact on the frequency of recorded side effects. Chemotherapy, endocrine therapy, anti-HER2-directed immunotherapy, all of which have a side effect profile heavily overlapping with that of bisphosphonates, make it virtually impossible to differentiate the actual reasons for observed symptoms.

Reviewer #2: Dear editors,

The manuscript entitled “The adverse effects of bisphosphonates in breast cancer: a systematic review and network meta-analysis” describe through and comprehensive analyses of the toxicity associated with bisphosphonates in breast cancer, both in the adjuvant and in the advanced setting. Quantification of the risks from bisphosphonates is of value and may better clarify to the risk and benefit balance when discussing with the patient treatment options. While it seems that the authors have invested many efforts and hard work for this study, there several important limitations to their work.

Major limitations:

1. “Data were extracted from a trials registry website when possible. If results were not published in a trials registry, all published papers regarding a trial were examined for adverse data, and data was taken from the most recent paper that included adverse data”. Data reported in the registry website often lacking or inaccurate (BMJ Open. 2017 Oct 11;7(10):e017719. doi: 10.1136/bmjopen-2017-017719) The most acceptable method in meta-analyses to use either the data reported in the manuscripts or to conduct patient-level data. This is a major limitation of this study.

2. “When ‘serious’ and ‘non serious’ events were reported (for example, as on clinicaltrials.gov), these were recorded as Grades 3/4 and Grades 1/2 respectively.” This use is incorrect at the definition of serious AE is different from the grading of CTCAE.

3. Results of studies comparing bisphosphonates to placebo/ no treatment should to be pooled together with studies comparing bisphosphonates to other treatment (such as denosumab).

Additional comments:

1. Generally, the text thought its sections is too long and not concise. This has negative effect on the readability of the manuscript. Specifically, the abstract should be more concise and should be shorten (currently 500 words). Description of the results should be shorter and more concise. In the results sections there are descriptions of the rational to preform and describe the analyses- some of these descriptions do not improve the clarity of the text and can be omitted, others should be moved to the methods section.

2. In the last paragraph of the introduction (row 173) details of the search should not be elaborated (this should be in the methods section), only concise description for the purpose of the study and the rationale for conduction the study.

4. Row 181-184 should be deleted.

5. Classification to early/ metastatic

6. Numbering of figures should be edited by the order it is mentioned in the manuscript

7. Figure 1-it is not clear how many studies were included. In the text box in the bottom of the figure- there is 56 studies. Do the text-boxes from the left to this text box describe additional studies?

8. I could not find figure 2 in the text.

9. Along to text there are preceding questions- to improve the readability of the manuscript, I suggest to omit these questions.

10. For the subgroup analysis in the AZURE study- could the interaction between the subgroup for the significant findings (arthritis) could be calculated and presented? Also, for the forest plots- could the size of the square or the circle be in proportion to the number of patients in each group?

11. For figure 4- can interaction for between the treatment options be presented for the different adverse event. According to the text fever and headaches showed significant interaction, but by the figure, additional adverse events might be influenced as well by the type of treatment

Reviewer #3: Freeman and co-authors provide an extensive and detailed analysis of adverse events associated with bisphosphonates from RCTs. The information are not new but tha amount of data and the detailed analysis add strenght to available evdidence.

Reviewer #4: Thank you very much for this interesting research question

I do have some questions and some major concerns:

Identification of studies:

Please could you clarify: how did you select studies? Your search terms seem to exclude studies if adverse events have not been reported in the abstract. If this is the case, please could you revise your search,. not to miss potential relevant studies?

Some more details why 37 studies have been excluded would be helpful (see also AMSTAR2)

Why did you include only 56 out of 101 studies?

Reporting of AEs:

Please could you explain: What did you count: number of AEs or participants with at least one AE? The unit of analysis is key not to double-count some participants.

Also related to double-counting: similar wording for different AEs in one study (e.g as reported in study registries): how did you ensure not the same patient was counted twice?

In table 1 serious AEs and >= grade 3 are counted together, but both have different definitions and should not be mixed up. A SAE (e.g leading to death, leading to hospitalisation and so on) is not necessarily >= grade 3 and vice versa.

Comparator:

I suggest not to combine placebo/no treatment and denosumab in the control arm, as denosumab is known to have similar safety and efficacy data compared to bisphosphonates

Pair-wise comparisons:

I suggest to use the random-effects model, as the included trials are clinically heterogenous

Network meta-analysis

Please provide some details how you tested transitivity assumptions

ROB

Please explain in more detail why ROB assessment is not feasible for saftey data. Especially ROB 2.0 is developed to assess on outcome level, and also to assess for per protocol analysis

6. PLOS authors have the option to publish the peer review history of their article (what does this mean?). If published, this will include your full peer review and any attached files.

Reviewer #1: No

Reviewer #2: No

Reviewer #3: **Yes: **pierfranco conte

Reviewer #4: No

---

## [Author Response · Author response to Decision Letter 0]

26 Oct 2020

Reviewer #1: The authors undertook a meta-analysis in order to describe more comprehensively than until now available the side effect profile of bisphosphonates. While this is a subject worth investigating, both because of the frequent use of bisphosphonates in many established clinical indications and the lack of systematic data, the attempt is severely hampered by a number of factors:

(1) most important: As the authors admit themselves, “…the analysis is limited by the availability and quality of data on adverse events, and the potential for bias introduced by a lack of standards for reporting of such events...“. I am afraid this essentially precludes a meaningful analysis, because the “garbage in – garbage out” principle cannot be “healed”…

Some of this is represented in the wide range of side effect frequencies reported (e.g. p3) – it does not make sense to the clinician to tell him/her that e.g. flu-like symptoms are increased to “….7-88%....”

We have sympathy with the reviewers’ point of view – analysis is always problematic when the data are of such variable quality and, as the reviewer points out, we have tried not to shy away from this. However, the methods we used allow us to take the variability between studies into account and express that in the uncertainties (as again the reviewer points out), and the stability of relative risks between studies is encouraging: as we say, illustrating with the example of fever, “the odds ratios and risk differences are more homogeneous between studies, indicating an increased risk of fever for people given bisphosphonates. This suggests that variable recording or reporting quality does not substantially influence the estimates of the treatment effects on risk.” Additional reassurance is the results’ alignment with clinical expectations. Taken together, these do indicate that the analysis is not without a sound basis.

The reviewer points out that some credible intervals are too wide to be clinically useful (‘7-88%’). That specific example was a particularly wide one, with most others giving more useful ranges. 

We appreciate the limitations of this work, but patients have requested this information. We feel that it is important firstly to do the best analysis possible with the currently-available data in order to inform their decision-making; secondly, to provide a methodology and precedent for others to follow in the future when more trials are reported with adverse events in full (we have made all our extracted data available as supplementary information to make it much easier for future re-analyses); and finally to raise awareness of some of the issues with the current adverse events reporting (and how it has improved since the development of clinical trials registries).

(2) Clinically, it is more or less meaningless to analyze drug side effects in trials from setting as different as neoadjuvant, adjuvant, and metastatic. This structural error of the approach is another reason for the wide variation of results. Furthermore, this mixing everything together annihilates possibly meaningful results for the individual settings. E.g. “Fever” may be quite negligible for the metastatic patient but less than trivial for the one-time user in a neoadjuvant setting. Another example: Myalgia and arthralgia, which are anyway frequent side effects of many adjuvant drugs in breast cancer, may not play a “bisphosphonate-specific” role in that setting, but be extremely cumbersome in otherwise healthy osteoporosis of prostate cancer patients, or patients with triple-negative breast cancer who do not receive aromatase inhibitors.

In this paper we investigated the differences between adjuvant and metastatic use, as well as the potential impact of different concurrent drug regimes and factors such as age and menopausal status of the women involved to avoid simply ‘mixing everything together’. 

Our estimates are based on randomised comparisons between control (women taking cancer therapy drugs but excluding bisphosphonate) and experimental groups (women taking equivalent cancer therapies but with bisphosphonates), thus they represent ‘bisphosphonate-specific’ side effects, above and beyond the side effects of the cancer drugs they are taking. 

The paper includes meta-regression analyses that examine how side-effects vary according to metastatic status and menopausal status. While there were limited data available to contrast these subgroups, we did identify some effects that were different between metastatic and non-metastatic patients. We also analysed the side effect data from two large trials of bisphosphonates as preventatives in women at high risk of breast cancer (FIT and HORIZON) in order to compare our results with those for women of an equivalent demographic but not taking cancer drugs. We judged this extra analysis and data to be beyond the scope of the paper, but we show it below to illustrate that the key side effects (including myalgia and arthralgia) remain significant even in this setting: 

Adverse effect Odds ratio Risk difference

Nausea 1.95 (1.57, 2.42) 0.01 (0.01, 0.01)

Arthralgia 3.54 (2.74, 4.59)

 0.02 (0.01, 0.02)

Fever 10.11 (8.5, 12.01)

 0.1 (0.1, 0.11

Headache 3.5 (2.9, 4.21)

 0.03 (0.03, 0.04)

Influenza-like symptoms 5.86 (4.77, 7.19)

 0.05 (0.04, 0.05)

Myalgia 5.99 (4.59, 7.81)

 0.03 (0.02, 0.03)

Chills 7.71 (4.98, 11.95)

 0.01 (0.01, 0.02)

Dehydration 23.01 (1.36, 390.54)

 0 (0, 0)

Diarrhoea 2.41 (1.48, 3.92)

 0 (0, 0)

Dizziness 1.89 (1.28, 2.78)

 0 (0, 0.01)

Fatigue 3.37 (2.53, 4.49)

 0.01 (0.01, 0.02)

Hot flushes 2.7 (1.31, 5.6)

 0 (0, 0)

Loss of appetite 6.48 (2.92, 14.38)

 0 (0, 0)

Musculoskeletal pain/event 0 (0, 0)

 0.05 (0.05, 0.06)

Nasopharyngitis 3.4 (1.25, 9.23)

 0 (0, 0

Oedema 4.5 (1.52, 13.32)

 0 (0, 0)

Eye disorders (not cataract) 11.03 (2.59, 46.93)

 0 (0, 0)

Vomiting 12.35 (5.37, 28.42)

 0.01 (0, 0.01)

A fixed effects meta-analysis of the adverse events reported from two large trials of bisphosphonates in women without breast cancer

(3) The lack of individual patient data reduces the value of the meta-analysis approach – having them available from just one trial (AZURE) that somewhat fall out of the average range both by its patient selection, excessive dose, and negative results heavily limits the meaningfulness of the results described.

The individual patient data was obtained only to address the research questions of whether the side effects of bisphosphonates vary significantly depending on the demographics of the women taking them (such as age and menopausal status) or concurrent therapies. AZURE was a large enough trial to be able to allow us to investigate these two questions from its data. The AZURE authors themselves state “combining ZOL with chemotherapy has the potential for enhanced toxicity, especially if there is a synergistic interaction with chemotherapy on normal tissues. Safety evaluation within the AZURE trial is the ideal opportunity to assess this. Here, we report the largest data set evaluating safety of ZOL outside the metastatic setting and the first analysis at this intensive dosing schedule addressing specific adverse events of note”. We therefore felt that this trial was an appropriate source of data for assessing the specific questions we were addressing.

The AZURE data do not excessively influence the meta-analysis results, compared to studies of similar size from which we did not have individual data, since all studies are included in the meta-analysis in the form of aggregate counts of events by treatment arm.

(4) As a result of mixing together many different clinical settings, indications, and stages, concomitant medication of patients must significantly impact on the frequency of recorded side effects. Chemotherapy, endocrine therapy, anti-HER2-directed immunotherapy, all of which have a side effect profile heavily overlapping with that of bisphosphonates, make it virtually impossible to differentiate the actual reasons for observed symptoms.

Our meta-analyses include only randomised trials. Therefore, the effects we estimate can interpreted as causal effects of bisphosphonates – either through including only comparisons of bisphosphonates and control (in the pairwise meta-analysis) or through the standard assumption of transitivity of relative treatment effects in the network meta-analysis.

Furthermore, the analysis of the AZURE data was designed to examine ways in which the effect of bisphosphonates might be modified by other factors. We examined how bisphosphonate effects varied according to concomitant medications, by estimating rates of symptoms within subgroups of the AZURE data defined by whether patients were taking endocrine therapy, chemotherapy or both (Figure 4). Since patients in AZURE were randomised to bisphosphonates or control, the observed difference between bisphosphonates and control within each of those concurrent therapy subgroups can be attributed to side effects of bisphosphonates, rather than the concurrent therapy.

Reviewer #2: Dear editors,

The manuscript entitled “The adverse effects of bisphosphonates in breast cancer: a systematic review and network meta-analysis” describe through and comprehensive analyses of the toxicity associated with bisphosphonates in breast cancer, both in the adjuvant and in the advanced setting. Quantification of the risks from bisphosphonates is of value and may better clarify to the risk and benefit balance when discussing with the patient treatment options. While it seems that the authors have invested many efforts and hard work for this study, there several important limitations to their work.

Major limitations:

1. “Data were extracted from a trials registry website when possible. If results were not published in a trials registry, all published papers regarding a trial were examined for adverse data, and data was taken from the most recent paper that included adverse data”. Data reported in the registry website often lacking or inaccurate (BMJ Open. 2017 Oct 11;7(10):e017719. doi: 10.1136/bmjopen-2017-017719) The most acceptable method in meta-analyses to use either the data reported in the manuscripts or to conduct patient-level data. This is a major limitation of this study.

The cited paper about inaccuracies on the registry website is solely about recruitment status (i.e. whether the registry is up to date with changing information). Becker et al (2014) found 20% of primary end points that could be compared between the registry and journal articles were discordant, with 14% of comparable secondary end points (doi:10.1001/jama.2013.285634). Pradhan et al (2019) automatically extracted data from clinicaltrials.gov and found that that data matched 87% with that extracted from journal papers (https://doi.org/10.1016/j.jclinepi.2018.08.023). Neither group drew conclusions over which source was more likely to be accurate. Hartung et al (2014) also looked at concordance between clinicaltrials.gov and journal publications, and specifically included adverse events. They found that out of 110 trials they examined, ‘Thirty-eight trials inconsistently reported the number of individuals with a serious adverse event (SAE); of these, 33 (87%) reported more SAEs in ClinicalTrials.gov. Among the 84 trials that reported SAEs in ClinicalTrials.gov, 11 publications did not mention SAEs, 5 reported them as zero or not occurring, and 21 reported a different number of SAEs.” And concluded “Which source contains the more accurate account of results is unclear, although ClinicalTrials.gov may provide a more comprehensive description of adverse events than the publication.” (https://doi.org/10.7326/M13-0480).

Our own experience (in most cases we checked all journal articles plus the clinical trials registry for each trial cited) matched that of Hartung et al. With adverse event data, unlike primary end point data, the reporting within journal articles is almost always far more limited than on clinicaltrials.gov. We also noted that different journal articles about the same trial often had conflicting information between them about the adverse effects. Sometimes this could be explained by longer time periods over which data had been collected (i.e. that some papers were early reports before trial completion), but not always, which implied that the journal articles were not a completely reliable source. Therefore, given the difficulty in determining which is more likely to be accurate in the case of discordance and the fact that the trial registry was a much fuller source of data, it was the one that we deferred to when it was available.

2. “When ‘serious’ and ‘non serious’ events were reported (for example, as on clinicaltrials.gov), these were recorded as Grades 3/4 and Grades 1/2 respectively.” This use is incorrect at the definition of serious AE is different from the grading of CTCAE.

We agree, and we have now cited in Table 1 the percentage of each adverse event classified as ‘serious’ in only the trials that reported ‘serious’ and ‘non serious’ adverse effects, to avoid mixing these two classification systems.

3. Results of studies comparing bisphosphonates to placebo/ no treatment should to be pooled together with studies comparing bisphosphonates to other treatment (such as denosumab).

In our pairwise meta-analysis, denosumab studies were excluded. This has now been clarified in the paper: “A standard fixed-effects meta-analysis was performed for each symptom independently, using studies that directly compared a bisphosphonate with a placebo or observation-only control”. 

In our network meta-analysis, the results of comparisons to placebo or observation-only are not pooled with comparisons to denosumab. Denosumab is treated as a separate comparator in the network. This allows the effects of comparisons with placebo, observation or denosumab to be distinguished, while allowing the data from the studies comparing bisphosphonates with denosumab to strengthen the evidence on absolute outcomes with bisphosphonates. 

Additional comments:

1. Generally, the text thought its sections is too long and not concise. This has negative effect on the readability of the manuscript. Specifically, the abstract should be more concise and should be shorten (currently 500 words). Description of the results should be shorter and more concise. In the results sections there are descriptions of the rational to preform and describe the analyses- some of these descriptions do not improve the clarity of the text and can be omitted, others should be moved to the methods section.

We agree. We had previously been asked to put numerical estimates for all outcomes in the Abstract, but have removed these again for readability, and tightened the wording of it whilst staying within reporting guidelines for Abstracts.

We have shortened the Results section and moved some of the material to Methods as suggested.

2. In the last paragraph of the introduction (row 173) details of the search should not be elaborated (this should be in the methods section), only concise description for the purpose of the study and the rationale for conduction the study.

We have cut this paragraph down.

4. Row 181-184 should be deleted.

The PRISMA reporting guidelines requested pre-registration information, but at the request of the reviewer we have removed it.

5. Classification to early/ metastatic

We’re not sure what change the reviewer was suggesting here. We have chosen to use ‘non-metastatic’ as opposed to the term ‘early’ as we have found that the term ‘early breast cancer’ can be interpreted differently by different readers whilst ‘non-metastatic’ was more specific.

6. Numbering of figures should be edited by the order it is mentioned in the manuscript

The figure numbering appears to be correct, but Figure 2 appears to have appeared at the end of the figures by mistake. We will ensure the ordering of the files uploaded to the publisher platform is corrected in the next round.

7. Figure 1-it is not clear how many studies were included. In the text box in the bottom of the figure- there is 56 studies. Do the text-boxes from the left to this text box describe additional studies?

The text boxes detail the reasons why the 47 studies (out of the 101 identified) were not included in the final total – leaving only 56. We have redrafted the figure to make this clearer.

8. I could not find figure 2 in the text.

Figure 2 is mentioned in the ‘meta-analysis of trials’ section of the Results.

9. Along to text there are preceding questions- to improve the readability of the manuscript, I suggest to omit these questions.

We were asked to include these by a previous reviewer who thought that they improved readability by ensuring the reader knew which analyses pertained to which explicit research question. We think this was perhaps helpful in the Methods section, but have removed the explicit questions from the Results section to help conciseness.

10. For the subgroup analysis in the AZURE study- could the interaction between the subgroup for the significant findings (arthritis) could be calculated and presented? 

Also, for the forest plots- could the size of the square or the circle be in proportion to the number of patients in each group?

11. For figure 4- can interaction for between the treatment options be presented for the different adverse event. According to the text fever and headaches showed significant interaction, but by the figure, additional adverse events might be influenced as well by the type of treatment

The interaction effects for each subgroup have been calculated and presented in each of Figures 3 and 4, and the most notable ones referred to in the text as follows, for Figure 3

“…a possible increase in arthritis as a side effect reported for women under 50 compared to those over 50 (ratio of relative risks between age 50 or over and under age 50: 0.22 (0.05, 1.04)). Fever was also possibly more likely in women with ER positive tumours (ratio of relative risks between ER positive and ER negative: 1.30 (0.88, 1.94))…”

And for Figure 4, “women not receiving chemotherapy were recorded as suffering slightly more fever (ratio of relative risks 4.8 (1.4, 16.1)).” and the mention of headache has been removed. 

The point sizes in Figures 3 and 4 are now scaled to emphasise the larger groups. 

Reviewer #3: Freeman and co-authors provide an extensive and detailed analysis of adverse events associated with bisphosphonates from RCTs. The information are not new but tha amount of data and the detailed analysis add strenght to available evdidence.

Thank you!

Reviewer #4: Thank you very much for this interesting research question

I do have some questions and some major concerns:

Identification of studies:

Please could you clarify: how did you select studies? Your search terms seem to exclude studies if adverse events have not been reported in the abstract. If this is the case, please could you revise your search,. not to miss potential relevant studies?

Our initial search terms included “(toxicity OR adverse OR side effect OR safety OR efficacy)” meaning that we were finding all trials in which either adverse effects or efficacy was mentioned in the abstract. We were in this search finding review articles as well as individual trials. As Figure 1 shows, this search identified 20 systematic reviews (or similar), from which trials were extracted as well as those caught in our initial search.

Some more details why 37 studies have been excluded would be helpful (see also AMSTAR2)

Why did you include only 56 out of 101 studies?

47 studies were excluded, and the reasons why are shown in Figure 1 (37 did not report adverse data, 2 trials were withdrawn without reporting, 4 have not yet reported results and 2 were reported in such a way as not to enable extraction of the adverse data for the relevant arms/patient subgroups separately). We have redrafted this figure to make it clearer that this is what is meant. Additionally, in the supplementary data, there is a complete list of all 101 trials and their characteristics with reasons for inclusion/exclusion. 

Reporting of AEs:

Please could you explain: What did you count: number of AEs or participants with at least one AE? The unit of analysis is key not to double-count some participants.

The number of participants with at least one of that type of AE (that is normally what is reported).

Also related to double-counting: similar wording for different AEs in one study (e.g as reported in study registries): how did you ensure not the same patient was counted twice?

We could not ensure this – as we state: “Some rarer events, such as many specific infections, cardiac symptoms (other than congestive heart failure) or non-breast cancer neoplasms, were collected together under a single heading. This means that for these events, if a single individual suffered 3 different kinds of infection, say, it would be recorded as 3 out of n experiencing ‘infection’.” However, none of these events came up as significant in the analysis, which would have prompted us to go back to the original data and re-extract them individually.

In table 1 serious AEs and >= grade 3 are counted together, but both have different definitions and should not be mixed up. A SAE (e.g leading to death, leading to hospitalisation and so on) is not necessarily >= grade 3 and vice versa.

We agree and have dealt with this by citing percentages in Table 1 only from studies which use the ‘serious’ versus ‘non serious’ classification of adverse events.

Comparator:

I suggest not to combine placebo/no treatment and denosumab in the control arm, as denosumab is known to have similar safety and efficacy data compared to bisphosphonates

In our pairwise meta-analysis, denosumab studies were excluded. This has now been clarified in the paper: “A standard fixed-effects meta-analysis was performed for each symptom independently, using studies that directly compared a bisphosphonate with a placebo or observation-only control”. 

In our network meta-analysis, the results of comparisons to placebo or observation-only are not pooled with comparisons to denosumab. Denosumab is treated as a separate comparator in the network. This allows the effects of comparisons with placebo, observation-only or denosumab to be distinguished, while allowing the data from the studies comparing bisphosphonates with denosumab to strengthen the evidence on absolute outcomes with bisphosphonates. 

Pair-wise comparisons:

I suggest to use the random-effects model, as the included trials are clinically heterogenous

Our primary meta-analysis results are from the network meta-analysis, which includes random effects, and fully represents the uncertainty arising from heterogeneity between treatment effects from different studies, the differences between classes of bisphosphonates, and incorporates direct and indirect evidence. To summarise the direct evidence alone, we used a simpler fixed effects meta-analysis, and we justified the use of fixed effects in the manuscript as follows:

“Note that fixed-effects meta-analysis still gives a valid estimate of a weighted average of the study-specific effects, even when, as in this analysis, there is heterogeneity between the study-specific estimates [30].”. 

Thus the fixed effects model validly estimates a quantity of interest without assuming homogeneity of effects. We think it is very unlikely that a classical random effects meta-analysis of the direct data alone would give any additional insights into our questions of interest.

Network meta-analysis

Please provide some details how you tested transitivity assumptions

The assumption of transitivity was tested by comparing direct evidence (e.g. A-C comparisons) with indirect estimates (e.g, from A-B and B-C comparisons). This was explained in the manuscript as follows: 

“In 15 out of the 157 treatment effects against observation-only from the best-fitting network meta-analysis models over all events, there was both direct and indirect evidence for the same treatment effect on the same event. In each case, the 95% credible limits for the direct vs indirect odds ratio spanned 1, indicating no important conflict between direct and indirect evidence.”

ROB

Please explain in more detail why ROB assessment is not feasible for saftey data. Especially ROB 2.0 is developed to assess on outcome level, and also to assess for per protocol analysis

Whilst many parts of ROB 2.0 could be applied to adverse event data (except Domain 5), as we stated, it would have ended up classifying all trials as at high risk of bias, because at least one of the domains would always have been at high risk.

For example, “3.1 Were data for this outcome available for all, or nearly all, participants randomized?” would have had to result in the answer ‘No Information’ for all trials, followed by answers to the follow-up questions that resulted in a high risk of bias rating for this domain (as patients who withdrew early from a study as a result of adverse events would have then had less potential for further adverse events and hence potentially biasing the sample results). Any single domain falling under ‘high risk of bias’ would result in an overall rating of ‘high risk of bias’. Hence, every trial would likely be classified as ‘high risk’ under this classification system.

A further concern was that ROB 2.0 did not include some aspects of potential bias that affect adverse event data specifically and hence could give an overall level of risk of bias lower than reality. For example, most trials specifically stated that (non serious) adverse event data was only collected for events where the clinician thought that the event could be related to the drug, and other trials almost certainly followed this same procedure even if they didn’t state it (this could be considered Domain 4?). There were also no classifications within ROB 2.0 related to potential conflicts of interest such as the funding source of the trial, which may influence the reporting of adverse events).

Our solution was to state the evidence for the domains that we felt helped differentiate the trials in terms of risk of bias. As stated in the manuscript, we considered the level of adverse event reporting (what percentage of patients had to develop a symptom before that symptom was reported as an outcome measure), the size of the trial, the blinding of the trial and (in the meta-data, released as supplementary information), the length of follow-up, the number of patients in each arm who withdrew due to adverse events, and the funding source of the trial. This, we hope, allows the readers to make some personal, at-a-glance assessment of the relative risks of bias between the studies.

We have additionally addressed the formatting, style and file naming conventions for both the main manuscript and the attached supplementary information.

---

## [Decision Letter · Decision Letter 1]

15 Jan 2021

PONE-D-20-18874R1

The adverse effects of bisphosphonates in breast cancer: a systematic review and network meta-analysis

PLOS ONE

Dear Dr. Freeman,

Thank you for submitting your manuscript to PLOS ONE. After careful consideration, we feel that it has merit but does not fully meet PLOS ONE’s publication criteria as it currently stands. Therefore, we invite you to submit a revised version of the manuscript that addresses the points raised during the review process.

I apologise for the delay in handling your manuscript, but apart from inherent difficulties of finding adept reviewers in this pandemic period, the reviewers' opinions varied vastly--for which reason I asked from another new reviewer to assist in this. Even then, the reviewers' suggestions ranged from rejection, to minor or major revision, to acceptance. I do believe the present submission is done in a systematic way and can certainly be of merit, despite its existing weaknesses. I therefore invite you to take into consideration the new comments provided by all reviewers and try to modify/improve the clarity and message of your submission, if possible.

We look forward to receiving your revised manuscript.

Kind regards,

Spyridon N. Papageorgiou, DDS, Dr Med Dent

Academic Editor

PLOS ONE

Reviewers' comments:

Reviewer's Responses to Questions

**Comments to the Author**

1. If the authors have adequately addressed your comments raised in a previous round of review and you feel that this manuscript is now acceptable for publication, you may indicate that here to bypass the “Comments to the Author” section, enter your conflict of interest statement in the “Confidential to Editor” section, and submit your "Accept" recommendation.

Reviewer #1: (No Response)

Reviewer #3: All comments have been addressed

Reviewer #4: (No Response)

Reviewer #5: (No Response)

2. Is the manuscript technically sound, and do the data support the conclusions?

Reviewer #1: No

Reviewer #3: Yes

Reviewer #4: No

Reviewer #5: Yes

3. Has the statistical analysis been performed appropriately and rigorously? 

Reviewer #1: N/A

Reviewer #3: Yes

Reviewer #4: No

Reviewer #5: Yes

4. Have the authors made all data underlying the findings in their manuscript fully available?

Reviewer #1: Yes

Reviewer #3: Yes

Reviewer #4: Yes

Reviewer #5: Yes

5. Is the manuscript presented in an intelligible fashion and written in standard English?

Reviewer #1: Yes

Reviewer #3: Yes

Reviewer #4: Yes

Reviewer #5: Yes

6. Review Comments to the Author

Reviewer #1: (No Response)

Reviewer #3: (No Response)

Reviewer #4: Thank you very much for your revised document, some of my comments are now clarified.

However, you did not reply to my comment:

Please provide some details how you tested transitivity assumptions

Your reply is related to consistency, not to transitivity, one main assumptions to conduct network meta-analysis (taking effect modifiers like different treatment duration, different observational time, different stages of disease and therefore different risks for the outcome of interest into account)

Reviewer #5: Thank you for clarifying the methods of this NMA.

I suggest one important Limitation that should be further discussed, in addition to the existing limitation that the trial investigators chose to report only those which they considered relevant.

We don't actually know how the monitoring was carried out i.e. did the trial investigators specifically ask closed questions on those AE of interest? Or was there a validated questionnaire used that sought to elicit particular symptoms in the same way across the trials? Equally, these symptoms may have been captured through spontaneous or open questions, and the investigators have filtered certain ones to upload to web registers or manuscripts.

So, selective monitoring or capture could have occurred, as did selective submission and analysis of AE of interest for reporting.

This means that some AE could have been over-emphasised, whilst others are missed.

Obviously this a common but important limitation of any meta-analysis of AE data.

7. PLOS authors have the option to publish the peer review history of their article (what does this mean?). If published, this will include your full peer review and any attached files.

Reviewer #1: No

Reviewer #3: No

Reviewer #4: No

Reviewer #5: No

---

## [Author Response · Author response to Decision Letter 1]

18 Jan 2021

Responses to the reviewers

Reviewer #4: Thank you very much for your revised document, some of my comments are now clarified.

However, you did not reply to my comment:

Please provide some details how you tested transitivity assumptions

Your reply is related to consistency, not to transitivity, one main assumptions to conduct network meta-analysis (taking effect modifiers like different treatment duration, different observational time, different stages of disease and therefore different risks for the outcome of interest into account)

*Thank you for your comments, and apologies if we did not respond appropriately to your query about transitivity. We understand "transitivity" as the assumption in network meta-analysis (NMA) that an A-C effect can be computed by combining data on A-B and B-C comparisons. It is fair to point out the issue of effect modification, since combining the data in this way is difficult to interpret if A-B studies and B-C studies involve populations with differing characteristics that modify the treatment effect. 

The events for which there was evidence of effect modification in our study are highlighted in Table 2 (increased bone pain, hypocalcaemia and fatigue, which varied between patients with and without metastases). For these events, the odds ratios estimated from the NMA (for all patients) are consistent with the corresponding odds ratio estimated from only direct comparisons, and the subgroup-specific estimates we present are based on direct data alone. Therefore we don't think that bias in the NMA due to effect modification affects our conclusions and that by presenting the data on the direct meta-analysis as well as the NMA should reassure readers of this. 

We have added a phrase to make this explicit at line 528, which now reads: “In each case, the 95% credible limits for the direct vs indirect odds ratio spanned 1, indicating no important conflict between direct and indirect evidence and providing reassurance about assumptions of transitivity.”

Reviewer #5: Thank you for clarifying the methods of this NMA.

I suggest one important Limitation that should be further discussed, in addition to the existing limitation that the trial investigators chose to report only those which they considered relevant.

We don't actually know how the monitoring was carried out i.e. did the trial investigators specifically ask closed questions on those AE of interest? Or was there a validated questionnaire used that sought to elicit particular symptoms in the same way across the trials? Equally, these symptoms may have been captured through spontaneous or open questions, and the investigators have filtered certain ones to upload to web registers or manuscripts.

So, selective monitoring or capture could have occurred, as did selective submission and analysis of AE of interest for reporting.

This means that some AE could have been over-emphasised, whilst others are missed.

Obviously this a common but important limitation of any meta-analysis of AE data.

*We entirely agree. We have added the sentence “There is also likely to be heterogeneity between trials in how adverse events were monitored and reported.” at line 754-755.

---

## [Editor Report · Decision Letter 2]

20 Jan 2021

The adverse effects of bisphosphonates in breast cancer: a systematic review and network meta-analysis

PONE-D-20-18874R2

Dear Dr. Freeman,

We’re pleased to inform you that your manuscript has been judged scientifically suitable for publication and will be formally accepted for publication once it meets all outstanding technical requirements.

Kind regards,

Spyridon N. Papageorgiou, DDS, Dr Med Dent

Academic Editor

PLOS ONE
---

## [Editor Report · Acceptance letter]

27 Jan 2021

PONE-D-20-18874R2 

The adverse effects of bisphosphonates in breast cancer: a systematic review and network meta-analysis 

Dear Dr. Freeman:

I'm pleased to inform you that your manuscript has been deemed suitable for publication in PLOS ONE. Congratulations! Your manuscript is now with our production department. 

Kind regards, 

on behalf of

Dr. Spyridon N. Papageorgiou 

Academic Editor

PLOS ONE